

# Spatially resolved dendritic integration: towards a functional classification of neurons

Christoph Kirch[1,2] and Leonardo L. Gollo[1,2,3]

[1] QIMR Berghofer Medical Research Institute, Brisbane, QLD, Australia
[2] Queensland University of Technology, Brisbane, QLD, Australia
[3] Turner Institute for Brain and Mental Health, School of Psychological Sciences, Monash University, Melbourne, VIC, Australia

## ABSTRACT

The vast tree-like dendritic structure of neurons allows them to receive and integrate input from many neurons. A wide variety of neuronal morphologies exist, however, their role in dendritic integration, and how it shapes the response of the neuron, is not yet fully understood. Here, we study the evolution and interactions of dendritic spikes in excitable neurons with complex real branch structures. We focus on dozens of digitally reconstructed illustrative neurons from the online repository NeuroMorpho.org, which contains over 130,000 neurons. Yet, our methods can be promptly extended to any other neuron. This approach allows us to estimate and map specific and heterogeneous patterns of activity observed across extensive dendritic trees with thousands of compartments. We propose a classification of neurons based on the location of the soma (centrality) and the number of branches connected to the soma. These are key topological factors in determining the neuron's energy consumption, firing rate, and the dynamic range, which quantifies the range in synaptic input rate that can be reliably encoded by the neuron's firing rate. Moreover, we find that bifurcations, the structural building blocks of complex dendrites, play a major role in increasing the dynamic range of neurons. Our results provide a better understanding of the effects of neuronal morphology in the diversity of neuronal dynamics and function.

# INTRODUCTION

Neurons are specialized excitable cells that are characterized by distinctive and often complex structures. Although the dendritic complexity is evident in various neuron types, it is often disregarded in computational models, and its role in dendritic integration in the presence of naturalistic stimuli is largely unknown. Moreover, the topology of dendritic trees may reflect fundamental elements for dendritic computation.

Each neuron in the brain is unique, and they can be classified into a myriad of neuron types and subtypes (*Masland, 2004*). Classification schemes often explore common properties of neurons, such as the neuron's morphology or neurotransmitter type (which can be excitatory or inhibitory), the anatomical region they belong to and their role in the

Corresponding author
Leonardo L. Gollo,
leonardo.gollo@monash.edu

circuit, or other dynamical and functional properties of neurons. With the ever-growing NeuroMorpho.org (*Ascoli, 2015*; *Ascoli, Donohue & Halavi, 2007*; *Halavi et al., 2012*) public online repository, there are more than 130,000 digital reconstructions of neuronal morphology available. These data can be used for anatomically realistic models (*Hines, Morse & Carnevale, 2007*; *Van Ooyen, 2011*; *Van Pelt, Van Ooyen & Uylings, 2001*) and morphometric analyses (*Wearne et al., 2005*). They can be invaluable for neuronal classification, which is typically based on morphology, electrophysiology, molecular, or functional properties (*Sharpee, 2014*; *Kanari et al., 2019*; *Wen & Chklovskii, 2008*). By obeying fundamental consistence rules such as enforcing neurons to have a tree topology (characterized by the absence of loops), digital reconstructions provide an unprecedented wealth of data with exquisite spatial resolution, which can be used to gain further insights on this complex problem of classifying and distinguishing different types of neurons.

In stark contrast to what some simple and influential point-neuron models suggest, such as the leaky integrate-and-fire model (*Brunel & Van Rossum, 2007*) and the Hodgkin–Huxley model (*Hodgkin & Huxley, 1952*), neurons can have large and intricate morphological structure that processes and integrates complex spatial patterns of input coming from thousands of synapses. Dendrites can filter electric input by leaking part of the current and propagate the rest. In addition to these passive properties, dendrites are also capable of producing supralinear amplification called dendritic spikes that occur owing to the voltage-gated dynamics of ion channels (*Häusser, Spruston & Stuart, 2000*; *Baer & Rinzel, 1991*; *Häusser & Mel, 2003*). These nonlinear and active properties of dendrites can boost the signal and generate interactions between neighboring compartments that are crucial to facilitate the transmission of information along extensive dendritic trees. Hence, spikes caused by neuronal integration of input from various synaptic sources depend on these non-linear (non-additive) dynamics taking place at dendrites with complex topology. Much work has been done to investigate how topology determines the capability of single neurons to detect intensity of stimulus (*Gollo, Kinouchi & Copelli, 2009*), to reliably detect dendritic spikes (*Schmidt-Hieber, Jonas & Bischofberger, 2007*), to discriminate input patterns (*De Sousa et al., 2015*), and to perform other forms of dendritic computation (*Koch & Segev, 2000*; *Sardi et al., 2017*; *Zang, Dieudonné & De Schutter, 2018*). There has also been some attempts to study this problem analytically (*Naud, Payeur & Longtin, 2017*; *Gollo, Kinouchi & Copelli, 2012*). However, given the complexity of the task, they are usually limited to regular or oversimplified dendritic structure (*Naud, Payeur & Longtin, 2017*).

Simple models play a major role at revealing fundamental dynamic mechanisms in neuroscience (*Avena-Koenigsberger, Misic & Sporns, 2018*). Here we describe the spatial structure of neurons and focus on main dynamics taking place at dendrites (*Segev & London, 2000*). Conventional multi-compartment models often have less than 100 compartments (*Van Ooyen et al., 2002*; *Remme, Rinzel & Schreiber, 2018*; *Shepherd et al., 1985*) and overlook the large number of synapses (about 10,000 in a human neuron) that lead to complex nonlinear interactions. Some approaches feature a detailed description of a specific neuron. However, a major limitation of this realistic approach is

the large number of parameters in the model (*Zang, Dieudonné & De Schutter, 2018*; *Keren, Peled & Korngreen, 2005*; *Mainen & Sejnowski, 1996*; *Eyal et al., 2018*; *Zandt, Veruki & Hartveit, 2018*; *Poirazi, Brannon & Mel, 2003*; *Hay et al., 2011*). Many of these parameters represent unknown variables, which is typical from such high-dimensional problems that include the description of dynamics of a variety of (often spatially dependent) ion channels along the dendritic tree.

Utilizing simplified neuronal dynamics, we mapped the heterogeneous response of dendritic compartments, independently subjected to stochastic excitatory input, in digitally reconstructed neurons. The structure of these neurons are trees that can be considered as complex networks, and consist of up to 10,000 compartments. We focused on the input-output response curve of these neurons as the intensity of incoming stimuli varies over several orders of magnitude. These response curves can be used to quantify essential features of the neuronal dynamics assuming a rate code (*Gollo, Kinouchi & Copelli, 2012*; *Kinouchi & Copelli, 2006*), as fundamental principles of neuronal functions may be determined by these dynamic features. Knowing the neuronal functions may be helpful as a way to classify and compare neurons and neuron types (*Carnevale et al., 1997*) within and across species (*Butler & Hodos, 2005*).

By applying this theoretical framework, our main aim is to investigate the implications of complex and realistic dendritic structure on dendritic integration and neuronal activity. We characterize the effects of topological properties of the neurons on the dynamic range of the response functions, which quantifies the ability of neurons to discriminate the intensity of incoming input, and show the contribution of bifurcations to spatially dependent activity. We identify distinctive dynamical behaviors of different types of neurons, induced by the dendritic topology, that reflect dynamical properties of the rate of activity of the soma with respect to the dendritic tree. Finally, we show how these findings can be explored to provide a novel functional classification of neurons, which is complementary to existing ones.

## METHODS

To estimate the spatial contribution of dendrites to the neuronal activity of digitally reconstructed neurons, a number of simplified assumptions was considered. The dynamics of each node was simulated using a simple model that represents the dynamics of excitable media. The dynamics of 26 different neurons from 6 different species (see details on Table 1), was characterized. Because details on the spatial distribution of all those neurons are not available, homogeneous dynamics was assumed, and a main focus was to better understand the contribution of the dendritic topology and their bifurcations on the dynamics of neurons. Given the large number of compartments and bifurcations that make up the dendritic arbor, any attempts of analytically modeling the propagation and interaction of potentially hundreds of spikes simultaneously are rendered nearly impractical and hence we focused on numerical experiments. Further, we expect these interactions to be highly non-linear owing to the heterogeneity of the neuronal topology. We overcome this complexity in spike dynamics by adapting a discrete computational model from previous studies (*Gollo, Kinouchi & Copelli, 2009*, *2012*, *2013*). Our model

**Table 1 List of neuron reconstructions used in this study, taken from NeuroMorpho.Org (version 7.6).**

| Label | Cell type | Region | Species | Number of somatic branches | Number of compartments | Number of bifurcations | Relative centrality of soma | NeuroMorpho ID | Reference |
|-------|-----------|--------|---------|------|------|------|------|------|------|
| A | Pyramidal | occipital; posteromedial visual, layer 5 | Mouse | 7 | 1,580 | 30 | 0.552 | NMO_72082 | D'Souza et al. (2016) |
| B | Pyramidal | frontal; primary motor | Human | 10 | 640 | 49 | 0.676 | NMO_84457 | Jacobs et al. (2018) |
| C | Pyramidal | anterior cingulate; layer 5 | Human | 7 | 1,441 | 213 | 0.756 | NMO_01058 | Watson, Jones & Allman (2006) |
| D | Pyramidal | temporal; Brodmann area 21, layer 2-3 | Human | 6 | 9,678 | 76 | 0.623 | NMO_68177 | Eyal et al. (2016) |
| E | Pyramidal | anterior cingulate; layer 5 | Human | 7 | 1,091 | 222 | 0.907 | NMO_01064 | Watson, Jones & Allman (2006) |
| F | Ganglion | retina; inner plexiform layer | Mouse | 6 | 3,936 | 50 | 0.827 | NMO_08168 | Mazzoni, Novelli & Strettoi (2008) |
| G | Pyramidal | parietal; Brodmann area 39 | Human | 6 | 413 | 24 | 0.943 | NMO_03500 | Jacobs et al. (2001) |
| H | Ganglion | retina; ganglion layer | Mouse | 5 | 2,876 | 51 | 0.956 | NMO_05321 | Coombs et al. (2006) |
| I | Pyramidal | frontal; primary motor, deep | Wallaby | 3 | 797 | 34 | 0.900 | NMO_84354 | Jacobs et al. (2018) |
| J | Sensory neuron | peripheral nervous system; cuticle | Drosophila melanogaster | 4 | 4,844 | 350 | 0.794 | NMO_79779 | Seco et al. (2017) |
| K | Pyramidal | hippocampus | Mouse | 3 | 775 | 38 | 0.741 | NMO_71409 | Bastian et al. (2017) |
| L | Pyramidal | medial prefrontal; layer 5 | Rat | 4 | 804 | 17 | 0.385 | NMO_66093 | Kougias et al. (2016) |
| M | Pyramidal | subiculum; stratum pyramidale | Rat | 5 | 856 | 20 | 0.325 | NMO_34951 | Routh et al. (2009) |
| N | Pyramidal | subiculum; stratum pyramidale | Rat | 6 | 708 | 17 | 0.352 | NMO_34958 | Routh et al. (2009) |
| O | Pyramidal | hippocampus; CA1; pyramidal layer | Mouse | 7 | 848 | 38 | 0.357 | NMO_50703 | Boillot et al. (2016) |
| P | Pyramidal | occipital; primary visual, layer 6 | Monkey | 9 | 751 | 28 | 0.336 | NMO_62656 | Briggs et al. (2016) |
| Label | Cell type | Region | Species | Number of somatic branches | Number of compartments | Number of bifurcations | Relative centrality of soma | NeuroMorpho ID | Reference |
|---|---|---|---|---|---|---|---|---|---|
| | | | | | | | | | |
| Q | Granule | hippocampus; dentate gyrus | Rat | 2 | 431 | 17 | 0.940 | NMO_00462 | *Rihn & Claiborne (1990)* |
| R | Purkinje | cerebellar cortex; vermis, Purkinje layer | Mouse | 2 | 5,726 | 358 | 0.879 | NMO_00864 | *Martone et al. (2003)* |
| S | Pyramidal | hippocampus; CA1; pyramidal layer | Rat | 2 | 473 | 47 | 0.500 | NMO_06145 | *Chapleau et al. (2009)* |
| T | Unknown | frontal | Rat | 2 | 511 | 22 | 0.327 | NMO_101378 | *Kuddannaya et al. (2018)* |
| U | Induced Neurons | forebrain | Human | 2 | 372 | 66 | 0.317 | NMO_103256 | *Bu et al. (2017)* |
| V | Purkinje | cerebellum; vermis, lobule III, apex | Mouse | 1 | 667 | 142 | 0.886 | NMO_80037 | *Jayabal, Ljungberg & Watt (2017)* |
| W | Interneuron | optic lobe; lobula complex; lobula plate | drosophila melanogaster | 1 | 4,659 | 762 | 0.826 | NMO_51008 | *Cuntz et al. (2013)* |
| X | Pyramidal | medial prefrontal; layer 2-3 | Rat | 1 | 636 | 15 | 0.698 | NMO_33937 | *Radley et al. (2013)* |
| Y | Purkinje | cerebellar cortex; Purkinje layer | Mouse | 1 | 1,568 | 129 | 0.382 | NMO_54509 | *Fukumitsu et al. (2016)* |
| Z | Purkinje | cerebellum; vermis, anterior, lobule V | Mouse | 1 | 993 | 107 | 0.196 | NMO_93863 | *Nedelescu, Abdelhack & Pritchard (2018)* |

preserves the main features of excitable systems, and by implementing real dendritic structures, we focus on the resultant spatial properties of neurons with active dendrites.

## Digital reconstructions

NeuroMorpho is a free online database of tens of thousands of three-dimensional neuron reconstructions. Each neuron has up to thousands of individual compartments, and the dataset is available in a standardized format, allowing the development of frameworks that can implement any neuron. In contrast to previous studies (*Gollo, Kinouchi & Copelli, 2009, 2012, 2013*), we take the entire spatial information provided by NeuroMorpho and treat the compartments as fundamental units of the neuron that are governed by identical dynamical rules (see "Compartment Dynamics"). We focused on a variety of neurons with high-quality reconstructions, based on visual inspection and diameter

regularity. The sampling was such that a large portion of a two-dimensional space comprising the number of branches connected to the soma and the relative centrality of the soma was covered, which were main topological features of neurons. More specifically, the list of neurons used in this paper is given in Table 1 (see below for a definition of the centrality). The NeuroMorpho version was 7.6.

## Compartment dynamics

We adapt a synchronous susceptible—infected (active)—refractory—susceptive model (SIRS) used in previous studies (Gollo, Kinouchi & Copelli, 2009). The model is a probabilistic cyclic cellular automata with discrete time and compartments, stochastic input, or deterministic evolution of the SIRS dynamics. A compartment switches states as a result of interactions from its neighboring compartments, stochastic input, or deterministic evolution of the SIRS dynamics. A compartment that is in the susceptible state (state 0) will remain there until activated either externally via synapses (see below), or by a propagation of activity from an active neighbor. A signal propagates to a susceptible neighboring compartment with a constant probability P. The probability of a failure to propagate (1−P) represents the net effect of two different contributions: the passive dumping of signal amplitude that propagates along dendrites, and the incoming synaptic inhibition. Both contributions can be responsible to prevent the threshold required to generate a dendritic spike to be reached, and thus represent a failure in propagation of activity. Once active (state 1), a compartment will switch to the refractory period (state 2) for a specific time, after which it will return to state 0. Here, we fix the refractory period to 7 time steps. Because of these dynamic rules, activity may spread to all susceptible neighboring compartment and travel in various directions (see Video S1). Moreover, two opposing signals will not add but annihilate each other (Royer & Miller, 2007). The model also recreates backpropagation, in which an action potential will travel back up the dendritic arbor once the soma has been activated (Waters, Schaefer & Sakmann, 2005). For simplicity, here we assume that the probability of forward and backward propagation is the same, as previous work incorporating different probabilities of propagation, depending on the direction, found that they affect the shape of the response function but have little influence on other measures such as the dynamic range (Gollo, Kinouchi & Copelli, 2009, 2012, 2013), which will be explored here. The soma itself also follows the same rules, however, it remains distinctive because it may be connected to many branches (Table 1). Previous works using regular dendrites have explored the effects of spatial-dependent dynamics (Gollo, Kinouchi & Copelli, 2013) and synaptic input (Gollo, Kinouchi & Copelli, 2009, 2012). Here, for simplicity, we assumed that the dynamics of all compartments is identical because detailed information regarding how heterogeneous activity takes place in various neuron types from different species and brain regions is absent.

In reality, many external factors are responsible for determining whether and when a compartment should fire. For example, a compartment could have thousands of synaptic connections, some inhibitory, some excitatory. In the end, however, the result will be either *On* (activate the compartment) or *Off* (remain in the susceptible state). We model this process using a probabilistic approach, where the probability of an excitatory synaptic signal is

$r = 1 - \exp(-h \cdot \delta t)$, where $h$ is the excitation rate, and $\delta t$ is the time step of 1 ms. Here we will focus on a range of $h$ that spans several orders of magnitude (from $10^{-4}$ to $10^4$ Hz), and $P$ that varies from 0.5 to 1 (as values of $P$ smaller than 0.5 exhibit a very strong attenuation, which is not plausible and give rise to little spatial contribution). The model is a discrete map, and the activation probability of a susceptible site $k_i$ neighbors, and can be written as:

$$P_i(t + \delta t) = 1 - (1 - r) \prod_{j=1}^{k_i} (1 - P)\delta(x_j, \ 1)$$

where $\delta(a, \ b)$ is the Kronecker delta, and $x_j$ is the state of the neighbor compartment $j$. We simulate each neuron for $10^6$ time steps (1,000 s) for combinations of $h$ and $P$, running every simulation five times. Over each simulation, we count the total number of times the soma fires ($F_S$) and the total number of times a dendritic compartment fires ($F_D$). The simulations were performed in MATLAB (MathWorks Inc., Natick, MA, USA) using a custom code (see "Code availability").

## Firing rate and dynamic range

The characteristic sigmoidal response function of a neuron is recovered by plotting the firing rate at some compartment against the excitation rate $h$ for some value of $P$. It shows how the neuron responds to various levels of input activity. An important feature of the response curve is the dynamic range, which represents the range of input rates that the neuron can effectively discern. By convention (*Kinouchi & Copelli, 2006*; *Gollo, 2017*), it is defined as:

$$\Delta = 10 \times \log_{10} \left( \frac{h_{90}}{h_{10}} \right)$$

where $h_{10}$ and $h_{90}$ correspond to the excitation rates that produce a firing rate that is 10% and 90% of the maximum firing rate.

## Relative energy consumption

Another metric of determining performance is the relative energy consumption, which we define here as

$$E = \frac{F_D/F_S}{N - 1}$$

where $N$ is the total number of compartments, and $F_D$ and $F_S$ are the number of dendritic and somatic spikes, respectively. The energy indicates how active the whole dendritic tree is compared to the soma, that is, how many times, on average, dendritic compartments activate for each somatic spike.

## Dynamics of benchmark neurites

To better understand the relationship between neuronal morphology and dynamics, we constructed and simulated a set of artificial neurites, which consist only of two branches. The primary branch is taken to be of fixed length $N$, whereas the secondary branch changes both in length $L$ and the position $Q$ at which it bifurcates from the primary

branch, where $Q$ is the index of the parent compartment of the primary branch. These toy neurons can be used to isolate the behavior we see in the full neurons.

## Centrality

A way of quantifying the soma's position in the neuron is by estimating how far away it is from the furthest endpoint. Let $T$ be the number of dendritic endpoints (branch terminals) of the neuron, and $D_{ij}$ be the distance (number of compartments) along the neuron from the $i$-th compartment to the $j$-th terminal compartment. Since no loops exist, this distance is always unique. Then we define the centrality of the $i$-th compartment as

$$C_i = \max_{j=1, \ldots, T} \{D_{ij}\}$$

Once the set of all compartmental centralities $C = \{C_1, \ldots, C_N\}$ has been calculated (excluding axonal compartments), the relative centrality of the soma is given by

$$C_{rel} = 1 - \frac{C_{soma} - \min\{C\}}{\max\{C\} - \min\{C\}}$$

where $C_{rel} = 0$ implies that the soma is the least central compartment of the neuron, while $C_{rel} = 1$ implies that the soma is the most central compartment.

## RESULTS

We investigate how morphology affects the dynamics of neurons. To focus on the topological properties of neurons with many compartments (100–10,000), we utilized a simple dynamic model: a canonical cyclic cellular automata model (*Gollo, Kinouchi & Copelli, 2009*; *Kinouchi & Copelli, 2006*; *Gollo, Copelli & Roberts, 2016*). We first introduce, illustrate, and characterize the relationship between neuronal structure and dynamics in a pyramidal mouse neuron (*D'Souza et al., 2016*). Then, we compare the resulting dynamical properties of multiple neurons from different species with a variety of neuronal cell types and structures (see "Methods"). Each compartment is considered to have synapses that can receive external input, and become active at a rate $h$, which is varied over several orders of magnitude. The activity propagates to quiescent (susceptible) neighbors with a probability $P$. By keeping track of the somatic and compartmental firing events, we quantify some fundamental dynamical properties of the neuron, such as the firing rate, relative energy consumption, and dynamic range. The emerging dynamics of the model is depicted in Fig. 1 (see also Video S1).

## Spatial maps of the activation rate

One strength of our computational model is the ability to simulate the complex dynamics of very large neurons with many compartments and including their specific digitally reconstructed morphology. To highlight the heterogeneity across dendritic branches, these results can be visualized spatially in the form of heat maps. The average rate of activation across time ($T = 10^6$ ms) of each compartment ($N = 1,580$) is illustrated in Fig. 2 for different values of $P$, the probability of propagation of activity (see Fig. S1 for other neurons). It is clear that the topology of the neuron affects the firing rate. Crucially, the

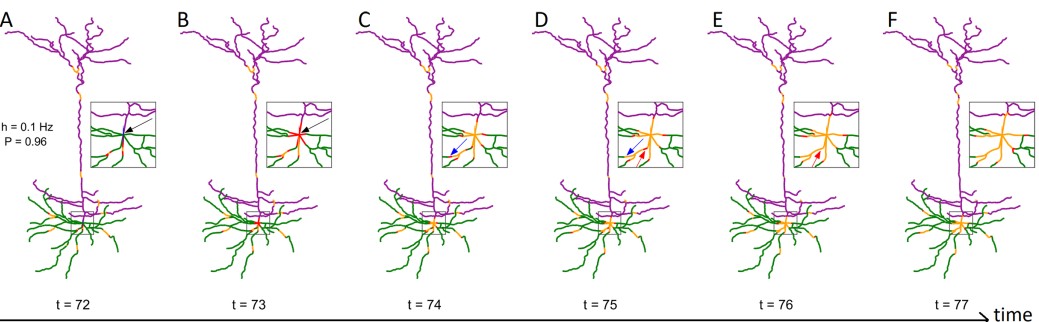

**Figure 1** **Series of snapshots showing how signals propagate along the neuron.** Visualization of digital reconstruction of neuron A (see "Methods"), where compartments are colored according to the following scheme: green for basal dendrites, purple for apical dendrites, blue for the soma (too small to be seen here), red for active (spiking) compartments, orange for refractory compartments. In each time step, spikes may propagate from active compartments to susceptible neighboring compartments with transmission probability $P$ (here, $P = 0.96$). A susceptible compartment may also spike due to the synaptic input, which we model stochastically with a Poisson rate of $h$ (here, $h = 0.1$ Hz). Once active, a compartment transitions to the refractory period and is unable to spike for 7 time steps. Panels (A–F) are different time points. The inset plot provides a closer look at the soma and surrounding compartments. (B) At $t = 73$, the neuron fires as a result of the dendritic integration of synaptic input (somatic spike, see black arrow). (C and D) Between $t = 74$ and $t = 75$, a signal fails to transmit (see blue arrow). (D and E) Between $t = 75$ and $t = 76$, two spikes can be seen annihilating each other (see red arrow). The snapshots were taken from Video S1.

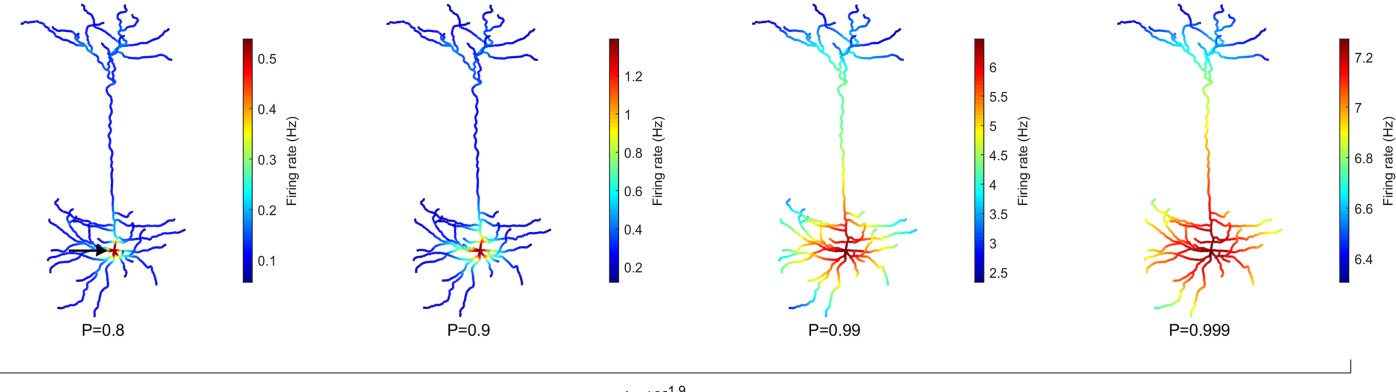

**Figure 2** **Spatial map of the compartmental firing rate.** The soma (marked by the arrow) lies among the most active region of the neuron, regardless of the signal transmission probability P. The amplification of the activity at the soma is stronger for lower values of $P$. Refer to Figs. S1 and S2 for heat maps of other neurons. Note different color scales.

soma (indicated by the arrow, left panel) becomes active at a higher rate than other compartments. Please note the different colormaps for the different panels. Because the model assumes homogeneity across compartments ($P$ and $h$), this amplification of firing rate at the soma occurs solely due to the topology. The soma has seven branches connected to it. These branches increase the likelihood of activations to reach the soma in comparison to other compartments because they can come from any active neighbor. Moreover, the lower $h$ and $P$ are, the greater is the relative amplification of the firing rate at the soma (see Fig. S2 for other neurons and values of $h$). These results also lead to the prediction that the activation rate can vary substantially across dendritic sites. For neuron

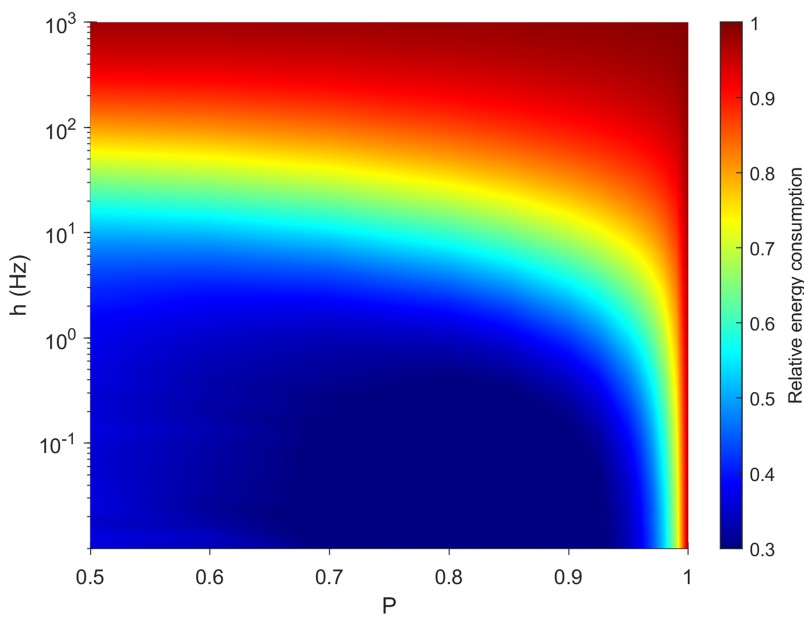

**Figure 3 Variations in relative energy consumption of the neuron over the parameter space.** The relative energy consumption of a neuron (demonstrated here using neuron A, see "Methods") is a measure of how often, on average, dendritic compartments spike per somatic spike. Here the energy consumption grows for high external driving and towards deterministic propagation ($P = 1$).

A (Fig. 2), the firing rate increases near the soma, but the firing rate is usually larger in sites close to a large number of bifurcations (see Figs. S1 and S2).

## Energy consumption for neuronal spikes

Active dendrites are capable of generating dendritic spikes, which allows for enhancement of activity and non-linear interactions. However, these dynamical benefits require additional energy. Here we estimated the relative energy consumption of the soma with respect to the number of activations the whole neuron experiences (see "Methods" for details). In our model, two processes are responsible for increasing the relative energy consumption of the neuron: (i) the external driving $h$, and (ii) the propagation and bifurcation of a dendritic spike as it passes a dendritic junction, initiating an additional spike that consumes energy. On the other hand, there are three mechanisms that reduce the relative energy consumption of the neuron: (i) energy dissipates stochastically due to the attenuation rate ($1-P$), (ii) signals can propagate in nonlinear waves that can be annihilated, and (iii) a signal traveling away from the soma will necessarily die when reaching an endpoint of a branch.

It is important for a neuron to balance its energy usage while performing its functions. In Fig. 3, we plot the relative energy consumption over the parameter space. The blue regions highlight areas of most efficient operation, as somatic spikes require fewer dendritic spikes to take place. In this case, it corresponds to a weak external driving rate ($h < 0.01$ Hz) and a rate of transmission that allows for failure of dendritic spike propagation ($P < 0.95$).

The behavior of our measure of energy consumption in the limits of the parameter space, as seen in Fig. 3, can be explained. Firstly, as $P$ approaches 1, every signal will be able to visit all compartments of the neuron once, or interact with another signal which will be able to visit the remaining compartments. In that case, the average firing rate of the compartments is identical and does not depend on topology. Hence, every dendritic compartment has to fire once for the soma to fire once (deterministic behavior). Secondly, in line with other studies (*Hasenstaub et al., 2010*), the energy consumption of spikes increases with $h$. Moreover, as $h$ approaches $\infty$, every compartment will fire independently at the highest possible rate (measured in hertz):

$$F_{\text{max}} = \frac{1}{(\text{duration of active state} + \text{duration of refractory state} + \delta t)}$$

Again, the average firing rate of the compartments is identical, hence we expect a relative energy consumption of 1. Only at these simple limiting cases is the dynamical behavior independent of morphology.

## Spatially resolved response function and dynamic range

One informative and influential way to quantify how dendritic trees process incoming signals is given by input-output response functions. It is defined by the mean output activation rate (across a long time interval, here $T = 10^6$ ms) as a function of the rate of activations induced by external driving $h$ (neuronal input). This means that the firing rate can be computed for each compartment. Response functions have their minimal in the absence of external input and their maximal for very strong external driving. As illustrated for different recording sites, response functions exhibit a sigmoidal shape (Fig. 4). These results show for this neuron a larger firing rate at the soma compared to other regions, especially at low external driving, which is consistent with the amplification of the firing rate observed at the soma (Fig. 2). At high values of $h$, the saturation of the response curves occurs in a similar manner regardless of the recording site. Moreover, for $P \approx 1$, this spatial heterogeneity vanishes.

An important feature of response functions that can be quantified corresponds to the dynamic range $\Delta$ (*Gollo, Kinouchi & Copelli, 2009*; *Kinouchi & Copelli, 2006*). It depends on the values of external driving at which the neuron responds at 10% of its maximal firing rate ($h_{10}$), and at 90% ($h_{90}$). The dynamic range quantifies the range between $h_{10}$ and $h_{90}$ (see "Methods" for details). It assumes that, based on the firing rate, the neuron is unable to reliably distinguish activation rates too close to saturation, $h < h_{10}$ and $h > h_{90}$. Figure 4 also illustrates the definition of the dynamic range for the response function measured at the soma and at a basal dendrite.

The dynamic range is a measure of the sensitivity to changes in the input rate. A large dynamic range indicates that a neuron can discern signals produced by a large range of input rates. For example, ganglion cells from the retina require a large dynamic range to be able to reliably respond to changes in lighting conditions that vary over several orders of magnitude (*Publio, Ceballos & Roque, 2012*).

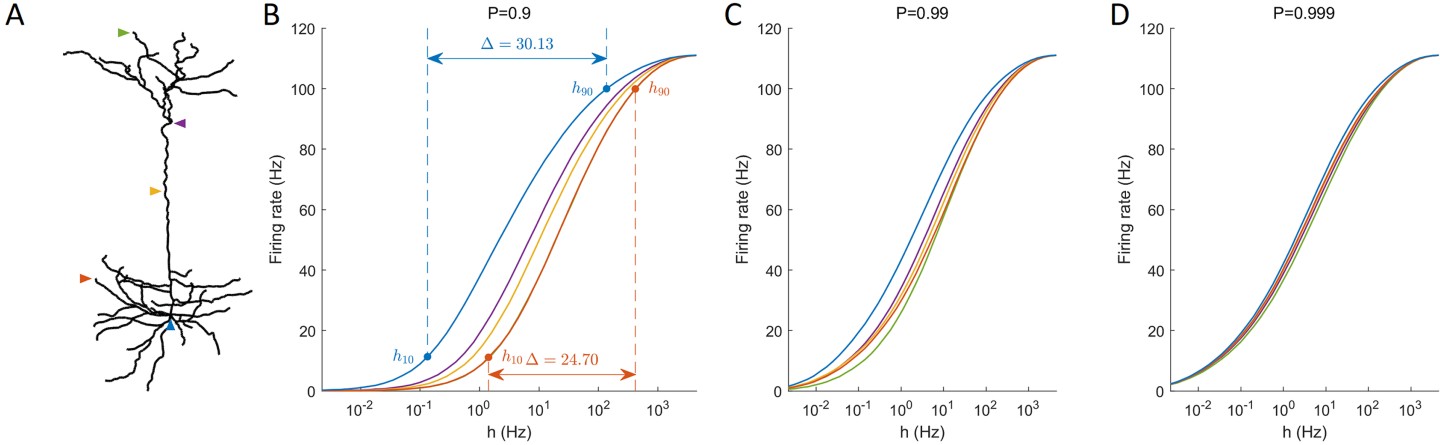

**Figure 4 Response functions at representative compartments.** (A) Marked compartments at different sites of the neuron. (B–D) Firing rate at different sites (marked in panel A) against the synaptic input rate $h$ for different values of $P$. (B) The dynamic range $\Delta$ is a function of $h_{10}$ and $h_{90}$ (see "Methods"). Lower values of $P$ yield stronger spatial dependance in the response functions of compartments.

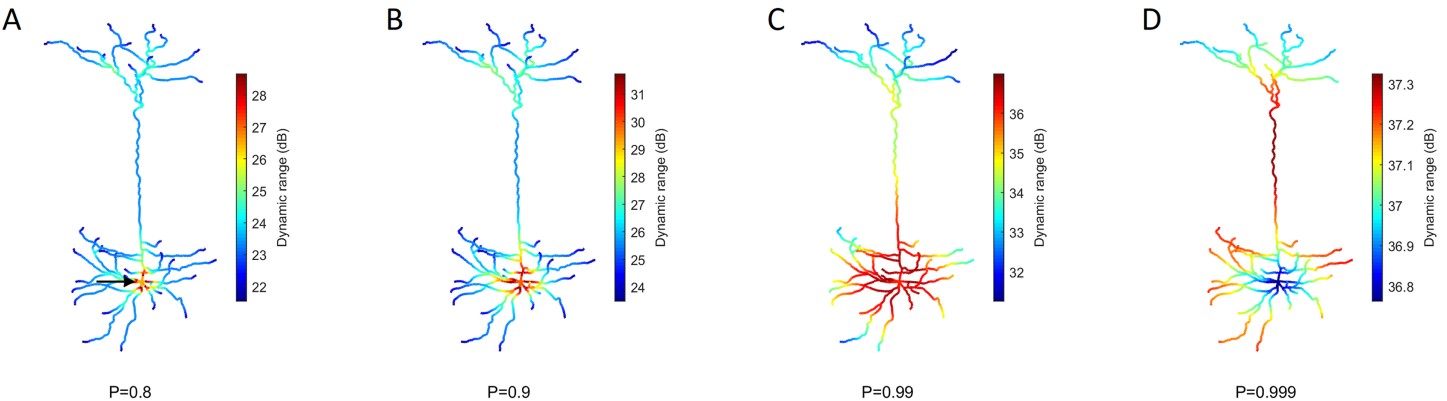

**Figure 5 Heat maps of the spatial distribution of the dynamic range.** (A–D) Dynamic range for different values of $P$. The soma (marked by the arrow) usually exhibits the opposite performance of the extremities. The dynamic range is larger at the soma for low transmission probability $P$, but lower for extremely high $P$. Please note the different color bars. See Fig. S3 for other neurons.

Our model allows us to identify the spatial gradients observed in the dynamic range (Fig. 5). Despite changes in the transmission probability $P$, the soma tends to exhibit high values of dynamic range. However, if $P$ is close enough to 1, the amplification of signals become detrimental, and the dynamic range at the soma can be lower than other regions. In this latter case, it is also relevant to notice that the differences in dynamic range across the neuron are overall very small (<1 dB). This happens because $h_{10}$ remains essentially unchanged whilst the saturation of the response function ($h_{90}$) occurs slightly earlier at the soma (see panel D of Fig. 4).

## Teasing apart the effects of a single branch on the dynamic range

To characterize the effects of neuronal topology, we explored the dynamic range of each compartment in the neuron as a function of its distance from the soma (Fig. 6). This new perspective reveals a general relationship that is mostly governed by $P$.

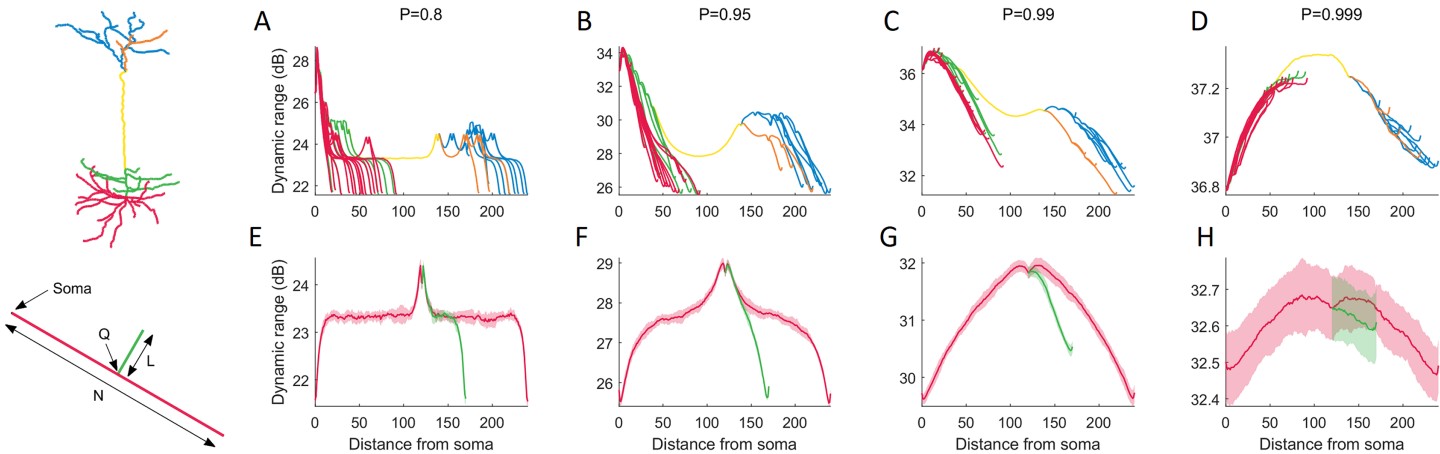

**Figure 6 Dynamic range as a function of the distance to the soma.** (A–D) The functional profile depends largely on the signal transmission probability P. Distinct structural regions of the neuron are color coded. For other neurons (see below), refer to Fig. S4. (E–H) Minimalistic structural model of a neuron with a main branch (red) of length N = 240, and a single bifurcation branch (green) of length L = 50. The secondary branch is connected to the primary branch at compartment index Q = 120. Many features in the variation of dynamic range can be exhibited by the simple toy neuron. The shaded regions are ±1 standard deviation of the mean, over 10 trials. The simple neurons were simulated for $10^5$ time steps. For additional results where we vary N, L and Q, refer to Fig. S5.

For $P < 0.99$, the dynamic range mostly decreases with the distance from the soma, and bifurcations generate a local boost in dynamic range, while a drop occurs at branch endpoints. For larger values of P, the dynamic range peaks at the main branch to the distal dendrites (yellow).

To better understand the relationship between dendritic topology and neuronal dynamics, we systematically studied how a single branch modifies the dynamic range. To pinpoint the effects of a single bifurcation on the dynamic range, we created a set of very simple neurons containing a single bifurcation with a small branch (Fig. 6E–6H). Starting with a primary branch (red) of constant length, we append a secondary branch (green) of length L to the primary branch at position Q, then run the simulations. Despite its simplified spatial structure, the minimal toy neuron faithfully reproduces many features in its dynamic range profile that we see from the full neuron reconstruction. For example, the effect of a single bifurcation or branch endpoint on the local dynamic range is consistent. A more complete set of toy neurites and their dynamic range is provided in Fig. S5.

## How does the dynamic range change across neurons?

Given the wide variety of neuronal morphologies, one might expect very different neurons to exhibit very different dynamics. In general, the dynamic range at the soma, the maximum dynamic range and the minimum dynamic range of the neuron increase with P (Fig. 7). At its highest, the dynamic range at the soma attains values of more than 35 dB for all neurons, and up to 43 dB. In addition, as shown before (Figs. 4 and 5), the measures of the dynamic range at the different sites become more homogeneous for very large values of P. The measure of the relative Δ at the soma shows that the soma is most often close to the sites of maximum dynamic range. However, some neurons
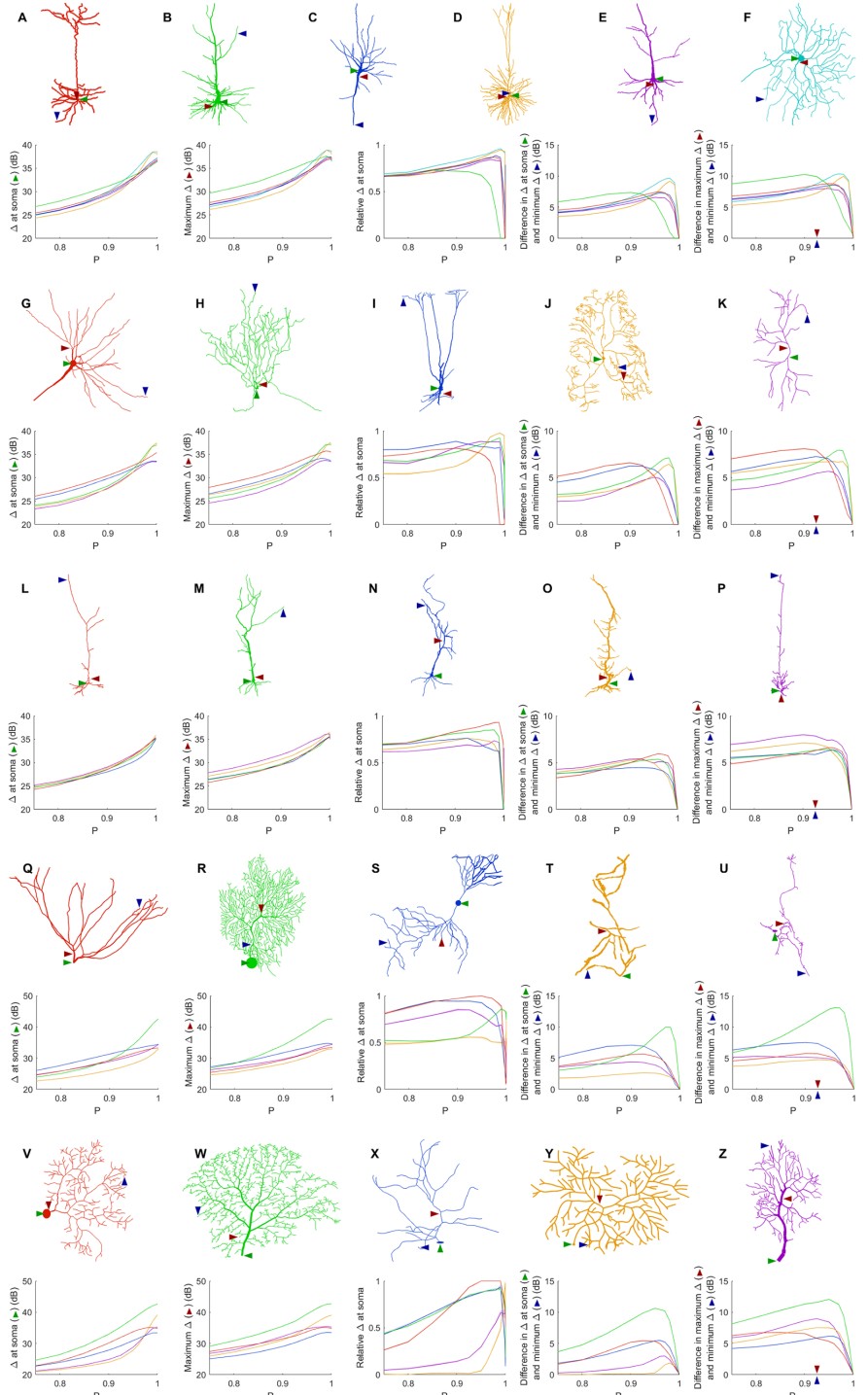

**Figure 7  Comparison of dynamic ranges across neurons.** Each row of graphs corresponds to the row of neurons above (see "Methods" for a description of each neuron). Colors are used only to differentiate the neurons of each row. The soma is marked by the green arrow, while the red and blue arrows indicate the location of the compartment at which the dynamic range is highest and lowest when $P = 0.92$. The relative dynamic range was calculated using $(\Delta_{soma} - \Delta_{min})/(\Delta_{max} - \Delta_{min})$. See Table 1 for original references and description of neurons.
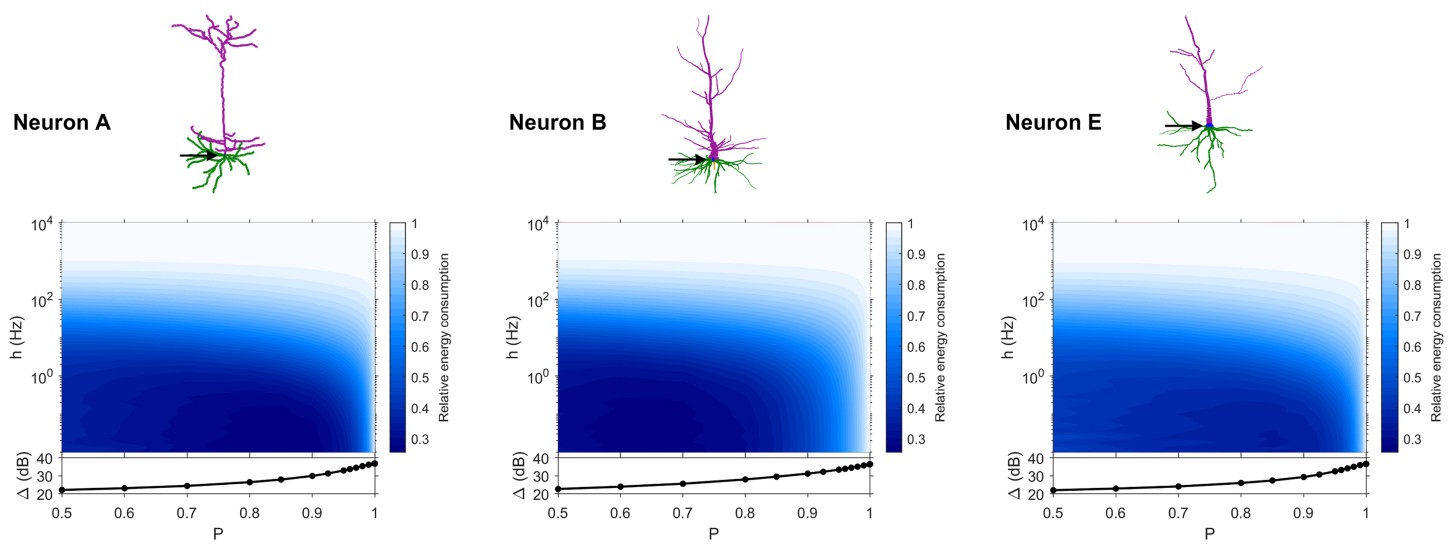

**Figure 8 Relative energy consumption for Type 1 neurons.** The location of the soma is indicated by the arrow, and its dynamic range is plotted in the bottom graph.

(R, T, Y and Z) exhibit the dynamic range at the soma somewhat smaller than the maximum dynamic range of the neuron. Moreover, the maximum heterogeneity of dynamic range across neurons varied substantially (from 7 to 13 dB). This spatial heterogeneity, even when the dynamics of the compartments is assumed to be identical, demonstrates that the topology of neurons plays a major role in shaping neuronal dynamics.

## Trends in energy consumption

The dynamic range tells us about the capability of neurons to encode stimuli that vary over orders of magnitude. However, this process has a cost, and the dynamic range does not reveal the neuronal efficiency in terms of energy consumption. Previously, we have introduced our measure of relative energy consumption, defined as the average number of times a dendritic compartment spikes for an action potential (somatic spike) to be generated (Fig. 3). If we compare the energy consumption across all neurons, three distinct types of behaviors emerge (Figs. 8–10), along with a transitioning behavior (Fig. 11). For the full set of energy consumption plots, refer to Fig. S6.

Type 1 (Fig. 8) occurs for the majority of neurons, despite the stark differences in morphologies. For these neurons, the energy is minimized for approximately $h < 10$ Hz and $P < 0.95$. Although the maximum dynamic range at the soma of every neuron occurs near $P = 1$, it corresponds to a very high relative energy consumption. However, a slight decrease in $P$ can almost minimize the energy consumption in these neurons, while keeping the dynamic range near its maximum. An optimally performing neuron would therefore slightly subject signal propagation to failure, saving energy without considerable loss in dynamic range.

Peer|

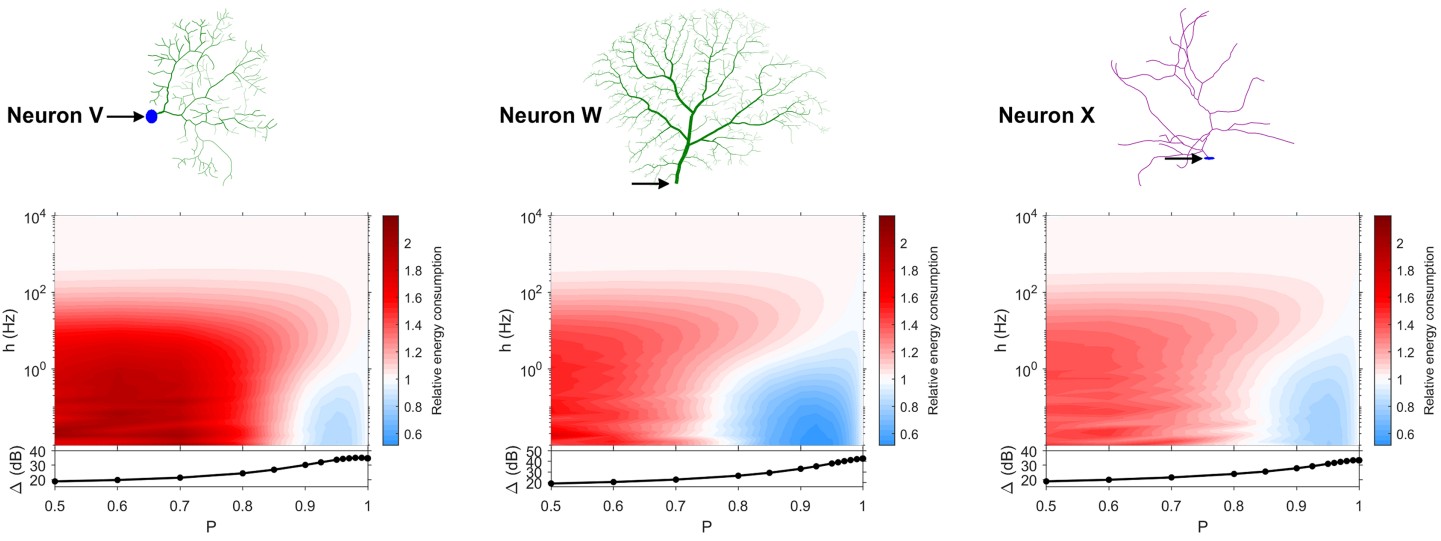

**Figure 9 Relative energy consumption for Type 2 neurons.** The location of the soma is indicated by the arrow, and its dynamic range is plotted in the bottom graph.

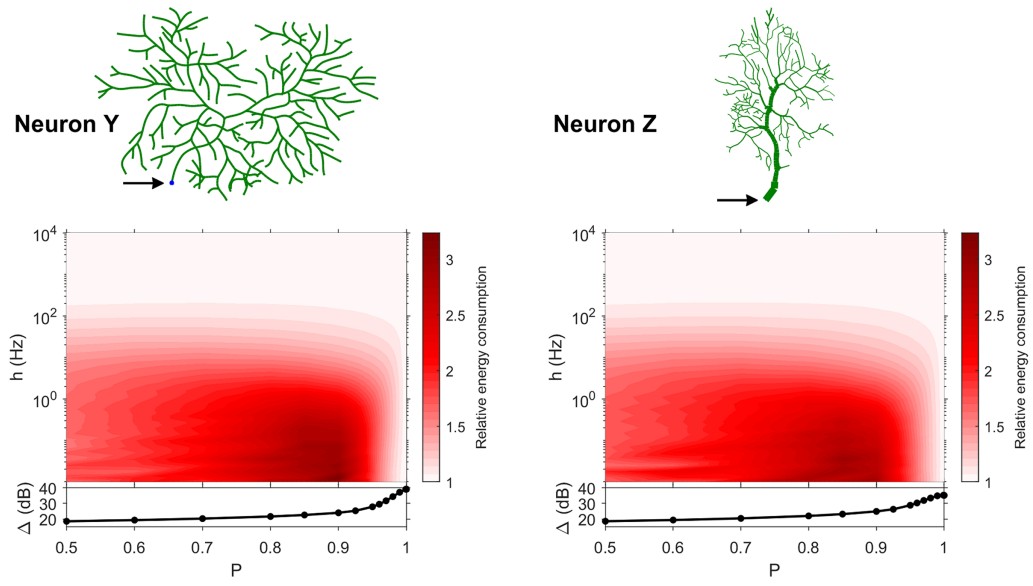

**Figure 10 Relative energy consumption for Type 3 neurons.** The location of the soma is indicated by the arrow, and its dynamic range is plotted in the bottom graph.

Type 2 (Fig. 9) corresponds to a narrower region of minimal energy consumption ($h < 1$ Hz and $0.85 < P < 0.95$). Unlike Type 1, decreasing the transmission probability below 0.8 is detrimental to the efficient operation of these neurons.

Type 3 (Fig. 10) display high relative energy consumption, with a maximum in the region $h < 1$ Hz and $0.8 < P < 0.9$. Moreover, the maximum and minimum energy consumption is much higher than for neurons of other types, and it never reaches a value below 1. As such, Type 3 seems to be intrinsically energy inefficient.

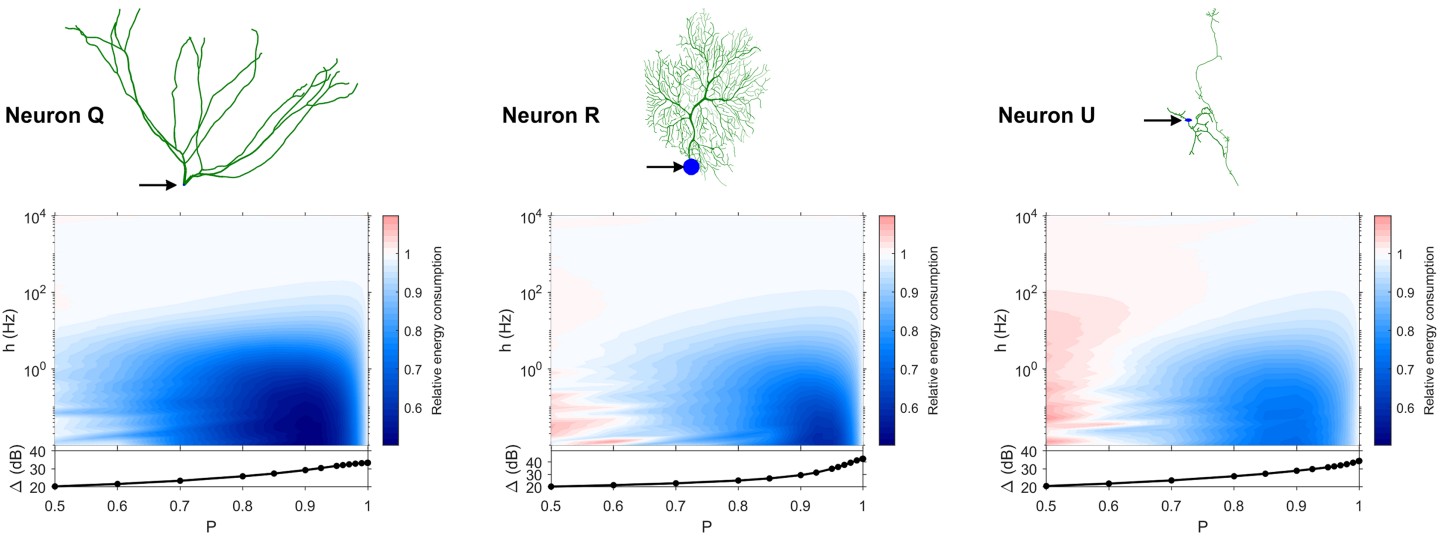

**Figure 11 Relative energy consumption for Type T neurons (transition type).** The location of the soma is indicated by the arrow, and its dynamic range is plotted in the bottom graph.                                 

The transition Type T (Fig. 11) exhibits a behavior that is between Types 1 and 2. Given the different behaviors, it is clear that the dendritic morphology affects the energy consumption of neurons. A crucial element in the computation of the energy consumption corresponds to the location of the soma, and the number of branches it has. Neurons of Type 1 have the soma located in a centralized position.

## Centrality

To quantitatively describe the soma's relative position within the overall extent of the dendritic arbor, we devise a general measure of centrality (see "Methods"). When $C = 1$, the soma is considered the most central compartment in the neuron. When $C = 0$, it is considered the least central. If a compartment is not central, it does not necessarily imply that it lies near the border of the neuron; for example, as for neuron A, compartments in the two separated regions of high bifurcation densities would experience a low centrality despite being surrounded by many compartments. Only the branch connecting these two areas would be central. Heat maps of the compartmental centrality for all neurons are provided in Fig. S7. The spatial mapping reveals that centrality is related to how symmetrically the rest of the neuron is distributed around a compartment.

## Categorization

Based on our estimation of energy consumption, the behavior of Type 2 can be distinguished from the behavior of Type 3 by the centrality (Fig. 12). Moreover, the behavior of Type 1 and the transition Type T can be explained by the number of branches connecting to the soma. This is an important property of neurons, as more branches naturally allow the soma to capture more information from the dendritic arbor. Then, if the soma is also located centrally, information can reach the soma more easily from any part of the neuron, and vice versa.

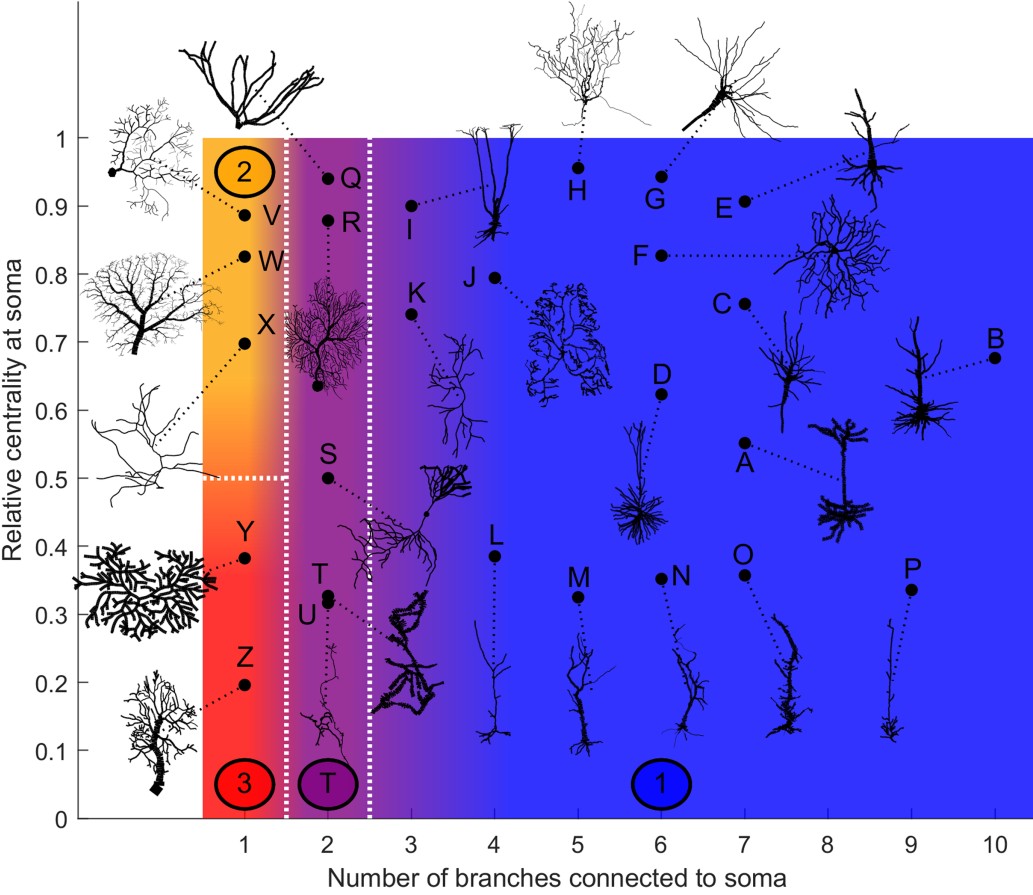

**Figure 12 Categorization of neurons according to their relative energy consumption profile.** Type 1 (blue) corresponds to Fig. 8, Type 2 (yellow) to Fig. 9, Type 3 (red) to Fig. 10, and Type T (transition, purple) to Fig. 11. The white dotted lines represent the approximate boundaries between different neuron types, and the colors indicate the different types.

Taking into account how these main structural features affect neuronal dynamics, we propose a categorization of neurons based on the relative centrality of the soma, and number of somatic branches (Fig. 12). Neurons in the same category exhibit qualitatively similar energy consumption profiles, and thus determine how efficiently the neuron operates over the parameter space: Type 1 neurons are intrinsically energy efficient, while Type 3 is intrinsically inefficient. The transition between classes is smooth. For example, neurons with 3 or 4 somatic branches have a higher minimum energy consumption than those with more somatic branches (see Fig. S6), however, they follow the same general behavior.

The general neuronal firing behavior of neurons can also be ascribed to their classification. For $P = 1$, the average response functions of neurons within types are very similar, as they do not depend much on the centrality and number of somatic branches. However, as $P$ decreases, each class tends to behave independently (Figs. 13A–13C). For a given input rate, neurons in category 1 fire more often, followed by the neurons in the transition regime, neurons of category 2 and neurons of category 3. Furthermore, we find

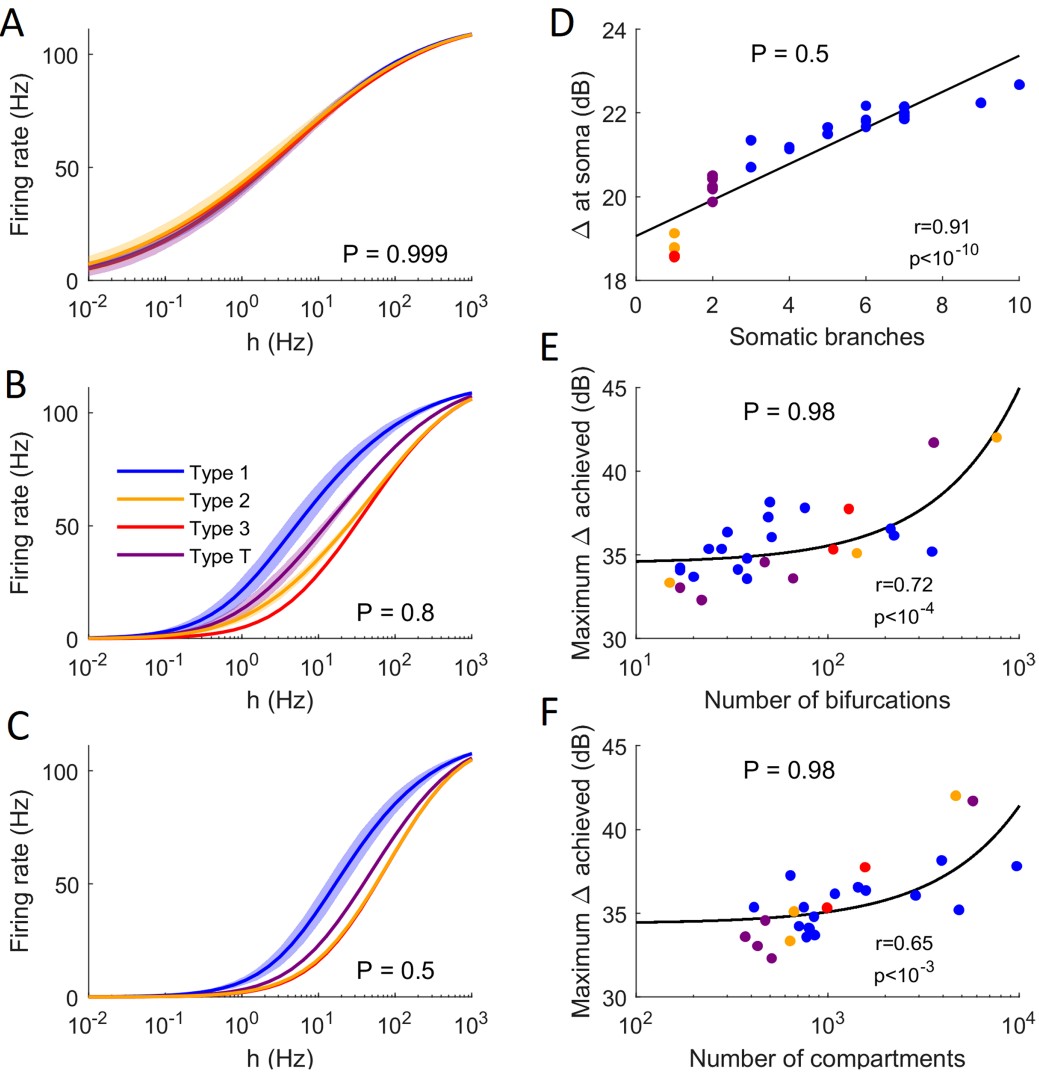

**Figure 13 Comparison of firing behavior.** (A–C) Average somatic response functions of the neurons in each category (see Fig. 12) for different values of *P*. The shading represents the standard deviation of firing rate within each group for a given stimulus intensity. For large *P*, all groups converge to the same behavior. For lower *P*, there is a distinct behavior between the average response functions of each group. (D) Dynamic range at the soma of each neuron against the number of somatic branches for *P* = 0.5. The colors represent the groups. The black trend line and Pearson correlation (*r* = 0.91) confirm that the dynamic range is strongly correlated to the number of somatic branches for lower values of *P*. (E and F) At high values of *P*, the number of bifurcations and number of compartments are good indicators of the maximum dynamic range that a neuron can achieve (here *P* = 0.98). Please note the logarithmic scale. The black trend lines and Pearson correlation show that the maximum Δ achieved is approximately linearly correlated with the neuron's size and complexity.

that a low probability of signal propagation has a detrimental effect on the somatic dynamic range of neurons with a low number of somatic branches (Fig. 13D). Additionally, neurons with a large number of compartments or bifurcations can achieve a higher maximum dynamic range independently of other morphological features

(Figs. 13E and 13F). How close this site with maximum dynamic range is to the soma, however, does depend on the neuronal morphology.

## DISCUSSION

Dendritic computation occurs as a result of multiple non-linear interactions taking place at dendrites (*Häusser, Spruston & Stuart, 2000*; *London & Häusser, 2005*). To provide insights into this phenomenon, we proposed a modeling approach that considers real neurons under naturalistic conditions, receiving independent synaptic-like input at thousands of compartments. These detailed neurons are spatially-extended excitable trees from NeuroMorpho, a database containing over 130,000 digitally reconstructed neurons (*Ascoli, 2015*; *Ascoli, Donohue & Halavi, 2007*; *Halavi et al., 2012, 2008*; *Donohue & Ascoli, 2011*; *Smith, Seligman & Swarup, 2008*; *Parekh & Ascoli, 2013*; *Ascoli et al., 2017*). Here we focus our results on a set of 26 selected neurons, however, it is possible to extend these analyses to any neuron of the database using the code provided.

We demonstrated the presence of substantial spatial dependence in neuronal dynamics that can be attributed to morphological features. Specifically, we mapped the excitability (firing rate) and dynamic range of dendritic branches and the soma. We identified bifurcations as a major structural source that can be very effective in raising the dynamic range. Furthermore, we showed how the number of branches connected to the soma and its centrality influence the energy consumption of neurons, and can be explored to classify neuron types. Hence, we classified neurons into three different families, based on centrality and number of branches connecting the soma, and a family that is within a transition zone. We found that a soma with only one branch is special, and a general behavior is expected when the soma has many branches. It is also possible to observe a transition that happens when the soma has two or three branches.

### Neuronal diversity

Diversity is a hallmark of neurons, and this is clearly demonstrated by the large variety of digitally reconstructed neurons found in the NeuroMorpho database, which currently has >130,000 neurons, from >640 cell types, and 60 species. Each of these neurons is unique. They have a tree topology, and their morphological features are crucial for classification. However, additional attempts have also been made to classify neurons based not only on their morphology but taking into account features of their electrophysiology, and their dynamics (*Masland, 2004*; *Sharpee, 2014*; *Markram et al., 2004*; *Mott & Dingledine, 2003*). These proposals attempt to improve neuronal classification with information about dynamics and function of neurons. Along this line, here we propose to incorporate a few key structural features that inform about neuronal dynamics and function. Our classification based on neuronal topology, together with other forms of neuronal classification that take into account species, anatomical region, morphological, and electrophysiological properties of neurons, may lead to more accurate functional classification schemes.

Utilizing a minimal dynamic model, we were able to simulate the dynamics of many neurons with thousands of compartments. This simple method is suitable to identify and

highlight the most important structural features of neurons. Here we focused on a variety of neurons, representing several neuron types, from different species, and acquired at multiple laboratories. Given this diversity, we did not focus on harmonizing the length of compartments. However, NeuroMorpho is a very rich dataset that allows a parcellation that forces compartments to have the same length in order to improve comparisons among neurons. The study was also primarily focused on neurons with high-quality and fine-resolution reconstructions (with large number of compartments), but it was comprised of a large proportion of pyramidal neurons. By incorporating more neuron types, the diversity of the sampling can be increased, and this approach may be effective to further explore the relationship between dendritic topology and neuronal function.

Action potentials and dendritic spikes consume energy because of the active flux of ions that is required to charge the membrane capacitance. To allow active signaling, these ions have to be pumped and this process uses energy provided by ATP. The energy cost of action potentials can vary considerably across neurons (*Sengupta et al., 2010*). Here we propose a different approach that focusses not on the cost of an action potential, but on the cost of a neuronal spike relative to the cost of the routing of electric activity through dendrites. We found that our estimation of relative energy consumption of neurons appears in stereotypical forms that can be linked to specific topological features. Hence, we propose that these features are relevant to characterize neuronal dynamics. According to our approach, it is clear that specific morphological fingerprints such as bitufted cells can be classified as belonging to a specific category (T). Furthermore, neurons with multiple branches are more effective at generating somatic spikes. These branches increase the convergence of input to the soma and reduce the overall density of dendritic spikes typically required to trigger a somatic spike. In contrast, neurons with a non-central soma connected to a single branch show the largest relative energy consumption, and they require more than one dendritic spike per compartment for each somatic spike. These distinct behaviors suggest specific computational function for neurons belonging to different families. Moreover, this simple approach reveals a clear role of dendritic topology, which might not be so evident in more complex neuron models that require a large number of parameters.

## Neuronal topology and dynamics

Other aspects of neuronal dynamics affected by dendritic topology include response function and dynamic range. It has been previously shown that the size of dendritic trees play a crucial role in determining the maximum dynamic range an active dendritic tree can attain (*Gollo, Kinouchi & Copelli, 2013*). However, this proposal was based only on a single topology (a regular and binary Cayley tree) with variable sizes (number of layers). This regular and artificial topology is relevant but the approach does not distinguish clearly the number of compartments from the number of bifurcations. Here, by utilizing real neurons from NeuroMorpho, we can assess the role of these two factors. We found that both the number of bifurcations (Pearson correlation, $r = 0.72$, $p < 10^{-4}$) as well as the number of compartments (Pearson correlation, $r = 0.65$, $p < 10^{-4}$) can be good

predictors for the maximum dynamic range of neurons. However, it must be taken into account that the number of compartments is also correlated with the number of bifurcations. In this dataset, it is important to consider that the number of compartments is mostly determined by the resolution of the digital reconstruction. In contrast, the number of bifurcations reflects more fundamental properties of dendritic topology.

Typically, the larger is the number of bifurcations, the larger is the dynamic range. This trend is also consistent with our minimal modeling approach containing a single bifurcation. We found that a single bifurcation tends to increase the dynamic range by a few decibels. A real dendritic topology can have many bifurcations that contribute to increase the dynamic range of the neuron. Although their contribution is not additive, it is reasonable to find that the number of bifurcations is perhaps the most important element to increase the dynamic range. As large dendrites have many bifurcations, the resolution used to digitalize such a complex structure needs to be fine. As a result, the number of compartments often correlates with the number of bifurcations. Together, our results suggest that the number of bifurcations is likely among the most essential features of dendrites to shape the dynamic range.

Centrality is also fundamental for the dynamics because central branches exhibit larger firing rates. This feature tends to increase the dynamic range as it is very effective at amplifying weak inputs. However, this amplification can also be beneficial to the dynamic range when $P$ approaches 1. In this case, when $P$ is nearly deterministic, we found that the dynamic range of central compartments is usually higher than at non-central ones (see Fig. S8). Going beyond single neurons, which have a tree topology with no loops, future work should test whether topological features (such as centrality and degree) of more general networks can also inform spatiotemporal patterns of activity in networks of larger scales of circuits, columns, and brain regions. Furthermore, the dynamics of networks depend on the diversity of neurons and properties of neuronal integration (*Gollo, 2017*; *Gollo, Copelli & Roberts, 2016*; *Gollo, Mirasso & Eguíluz, 2012*). Hence, it remains to be determined how the different types of neurons proposed here influence the activity of networks.

### Neuronal models

Our model is suitable to explore the topological effects of tens of thousands of digitally reconstructed neurons with thousands of compartments in neuronal dynamics under complex input conditions. In order to focus on these crucial spatial aspects of neurons, here we considered explicitly simplified neuronal dynamics. We disregarded heterogeneity of ion channels along neurons and assumed homogeneity for simplicity. This allows us to highlight topological features of neurons and ascribe the heterogeneous types of dynamics exclusively to dendritic structure. We argue that our simple modeling approach retains the essential features to simulate the dynamics of excitable systems without the burden of an excessive number of details and parameters. However, this is clearly a simplification, and future work should address the role of more sophisticated biophysical models with additional free parameters that describe the membrane potential of dendritic branches as continuous variables (differential equations, instead of a map with discrete

states (*Girardi-Schappo, Tragtenberg & Kinouchi, 2013*)), and explicitly consider the contributions of excitatory and inhibitory synapses. Here, as a first step, inhibition is only implicitly considered in the net synaptic contribution since in a previous study using a similar dynamic model inhibition did not show much impact in the dynamic range of the network whilst requiring additional free parameters (*Gollo, Copelli & Roberts, 2016*). More detailed models might also incorporate heterogeneity of dynamics, taking into account impedance gradients, the dendritic diameter, type of dendrite, distance from soma, and so on (*Donohue & Ascoli, 2008*). Future studies including electrotonic analysis will require more parameters but will represent an important validation step of our findings and may lead to a better understanding of the relationship between dendritic topology and function. Moreover, our simplifying assumptions of homogeneous and constant input can also be extended in future works. As a first step, a more detailed description of the dynamics of the soma can be considered in which the model has two types of dynamics, one for the dendrites and one for the soma, and the role of two types of integration can be independently assessed. This distinction might be relevant because it is known that signal integration at the soma can be crucial for coincidence detection and the resulting network response (*Gollo, Mirasso & Eguíluz, 2012*).

Changes in neuronal structure are reported in many neuropsychiatric disorders (*Uylings & De Brabander, 2002*; *Coleman & Flood, 1987*; *Kulkarni & Firestein, 2012*; *Raymond, Bauman & Kemper, 1995*; *Forrest, Parnell & Penzes, 2018*). The minimal modeling approach proposed here can be used to characterize the changes in neuronal dynamics caused by these structural alterations (*Kirch & Gollo, 2020*). Our simplified approach might also be relevant for simulating motifs and circuits of neurons with detailed dendritic structure under complex and realistic input conditions. Future work can focus on more complex and specific extensions of the model. For example, spatial- and time-dependent input may reveal other main features of neuronal dynamics with dendritic computation taking place in parallel at functional subunits in cortical circuits.

## CONCLUSIONS

Our model provides insights into the role the dendritic structure plays in the behavior of a neuron, both local and on the larger scale. We used digital reconstructions of real neurons to address the effects of intricate and nonhomogeneous spatial features of neurons on their dynamics. Our results indicate that two main morphological features—the centrality of the soma, and the number of branches connected to the soma—can determine the type of behavior a neuron exhibits. Neurons whose soma lies on the border and in a non-central location are intrinsically energy inefficient, whereas neurons with many branches connected to the soma are intrinsically energy efficient. Furthermore, we have shown that bifurcations in the dendritic tree can enhance the dynamic range, and that the maximum dynamic range of neurons increase with the number of bifurcations and compartments. Our approach can be extended to more than 130,000 neurons available at the NeuroMorpho database.

### Code availability

Matlab code to reproduce the results, named "Neurodynamics", is available at Systems Neuroscience Group: http://www.sng.org.au/Downloads.

### Funding

This work was supported by the Australian Research Council and the Australian National Health and Medical Research Council (APP1110975). The work was also supported by the Dentons Australia Honors Scholarship. The funders had no role in study design, data collection and analysis, decision to publish, or preparation of the manuscript.

### Grant Disclosures

The following grant information was disclosed by the authors:
Australian Research Council.
Australian National Health and Medical Research Council: APP1110975.
Dentons Australia Honors Scholarship.

### Competing Interests

Leonardo L. Gollo is an Academic Editor for PeerJ.

### Author Contributions

- Christoph Kirch conceived and designed the experiments, performed the experiments, analyzed the data, prepared figures and/or tables, authored or reviewed drafts of the paper, and approved the final draft.
- Leonardo L. Gollo conceived and designed the experiments, analyzed the data, authored or reviewed drafts of the paper, and approved the final draft.

### Data Availability

Matlab code to reproduce the results, named "Neurodynamics", is available at Systems Neuroscience Group: http://www.sng.org.au/Downloads.

### Supplemental Information

Supplemental information for this article can be found online at http://dx.doi.org/10.7717/peerj.10250#supplemental-information.

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

# PeerJ

**Jacobs B, Schall M, Prather M, Kapler E, Driscoll L, Baca S, Jacobs J, Ford K, Wainwright M, Treml M. 2001.** Regional dendritic and spine variation in human cerebral cortex: a quantitative golgi study. *Cerebral Cortex* **11(6)**:558–571.

**Jayabal S, Ljungberg L, Watt AJ. 2017.** Transient cerebellar alterations during development prior to obvious motor phenotype in a mouse model of spinocerebellar ataxia type 6. *Journal of Physiology* **595(3)**:949–966.

**Kanari L, Ramaswamy S, Shi Y, Morand S, Meystre J, Perin R, Abdellah M, Wang Y, Hess K, Markram H. 2019.** Objective morphological classification of neocortical pyramidal cells. *Cerebral Cortex* **29(4)**:1719–1735.

**Keren N, Peled N, Korngreen A. 2005.** Constraining compartmental models using multiple voltage-recordings and genetic algorithms. *Journal of Neurophysiology* **94(6)**:3730–3742 DOI 10.1152/jn.00408.2005.

**Kinouchi O, Copelli M. 2006.** Optimal dynamical range of excitable networks at criticality. *Nature Physics* **2(5)**:348.

**Kirch C, Gollo LL. 2020.** Dynamical effects of dendritic pruning implicated in aging and neurodegeneration: towards a measure of neuronal reserve. *BioRxiv* DOI 10.1101/2020.04.09.035048.

**Koch C, Segev I. 2000.** The role of single neurons in information processing. *Nature Neuroscience* **3(11s)**:1171.

**Kougias DG, Nolan SO, Koss WA, Kim T, Hankosky ER, Gulley JM, Juraska JM. 2016.** Beta-hydroxy-beta-methylbutyrate ameliorates aging effects in the dendritic tree of pyramidal neurons in the medial prefrontal cortex of both male and female rats. *Neurobiology of Aging* **40**:78–85.

**Kuddannaya S, Tong CS, Fan Y, Zhang Y. 2018.** Geometrically mediated topographic steering of neurite behaviors and network formation. *Advanced Materials Interfaces* **5(7)**:1700819.

**Kulkarni VA, Firestein BL. 2012.** The dendritic tree and brain disorders. *Molecular and Cellular Neuroscience* **50(1)**:10–20.

**London M, Häusser M. 2005.** Dendritic computation. *Annual Review of Neuroscience* **28**:503–532.

**Mainen ZF, Sejnowski TJ. 1996.** Influence of dendritic structure on firing pattern in model neocortical neurons. *Nature* **382(6589)**:363.

**Markram H, Toledo-Rodriguez M, Wang Y, Gupta A, Silberg G, Wu C. 2004.** Interneurons of the neocortical inhibitory system. *Nature Reviews Neuroscience* **5(10)**:793.

**Martone ME, Zhang S, Gupta A, Qian X, He H, Price DL, Wong M, Santini S, Ellisman MH. 2003.** The cell-centered database. *Neuroinformatics* **1(4)**:379–395.

**Masland RH. 2004.** Neuronal cell types. *Current Biology* **14(13)**:R497–R500.

**Mazzoni F, Novelli E, Strettoi E. 2008.** Retinal ganglion cells survive and maintain normal dendritic morphology in a mouse model of inherited photoreceptor degeneration. *Journal of Neuroscience* **28(52)**:14282–14292.

**Mott DD, Dingledine R. 2003.** Interneuron diversity series: interneuron research-challenges and strategies. *Trends in Neurosciences* **26(9)**:484–488.

**Naud R, Payeur A, Longtin A. 2017.** Noise gated by dendrosomatic interactions increases information transmission. *Physical Review X* **7(3)**:031045.

**Nedelescu H, Abdelhack M, Pritchard AT. 2018.** Regional differences in Purkinje cell morphology in the cerebellar vermis of male mice. *Journal of Neuroscience Research* **96(9)**:1476–1489.

**Parekh R, Ascoli GA. 2013.** Neuronal morphology goes digital: a research hub for cellular and system neuroscience. *Neuron* **77(6)**:1017–1038.

**Poirazi P, Brannon T, Mel BW. 2003.** Pyramidal neuron as two-layer neural network. *Neuron* **37(6)**:989–999.

**Publio R, Ceballos CC, Roque AC. 2012.** Dynamic range of vertebrate retina ganglion cells: importance of active dendrites and coupling by electrical synapses. *PLOS ONE* **7(10)**:e48517.

**Radley JJ, Anderson RM, Hamilton BA, Alcock JA, Romig-Martin SA. 2013.** Chronic stress-induced alterations of dendritic spine subtypes predict functional decrements in an hypothalamo-pituitary–adrenal-inhibitory prefrontal circuit. *Journal of Neuroscience* **33(36)**:14379–14391.

**Raymond GV, Bauman ML, Kemper TL. 1995.** Hippocampus in autism: a Golgi analysis. *Acta Neuropathologica* **91(1)**:117–119.

**Remme MW, Rinzel J, Schreiber S. 2018.** Function and energy consumption constrain neuronal biophysics in a canonical computation: coincidence detection. *PLOS Computational Biology* **14(12)**:e1006612.

**Rihn LL, Claiborne BJ. 1990.** Dendritic growth and regression in rat dentate granule cells during late postnatal development. *Developmental Brain Research* **54(1)**:115–124.

**Routh BN, Johnston D, Harris K, Chitwood RA. 2009.** Anatomical and electrophysiological comparison of CA1 pyramidal neurons of the rat and mouse. *Journal of Neurophysiology* **102(4)**:2288–2302.

**Royer AS, Miller RF. 2007.** Dendritic impulse collisions and shifting sites of action potential initiation contract and extend the receptive field of an amacrine cell. *Visual Neuroscience* **24(4)**:619–634.

**Sardi S, Vardi R, Sheinin A, Goldental A, Kanter I. 2017.** New types of experiments reveal that a neuron functions as multiple independent threshold units. *Scientific Reports* **7(1)**:18036.

**Schmidt-Hieber C, Jonas P, Bischofberger J. 2007.** Subthreshold dendritic signal processing and coincidence detection in dentate gyrus granule cells. *Journal of Neuroscience* **27(31)**:8430–8441.

**Seco CZ, Castells-Nobau A, Joo SH, Schraders M, Foo JN, Van der Voet M, Velan SS, Nijhof B, Oostrik J, De Vrieze E, Katana R. 2017.** A homozygous FITM2 mutation causes a deafness-dystonia syndrome with motor regression and signs of ichthyosis and sensory neuropathy. *Disease Models & Mechanisms* **10(2)**:105–118.

**Segev I, London M. 2000.** Untangling dendrites with quantitative models. *Science* **290(5492)**:744–750.

**Sengupta B, Stemmler M, Laughlin SB, Niven JE. 2010.** Action potential energy efficiency varies among neuron types in vertebrates and invertebrates. *PLOS Computational Biology* **6(7)**:e1000840.

**Sharpee TO. 2014.** Toward functional classification of neuronal types. *Neuron* **83(6)**:1329–1334.

**Shepherd G, Brayton R, Miller J, Segev I, Rinzel J, Rall W. 1985.** Signal enhancement in distal cortical dendrites by means of interactions between active dendritic spines. *Proceedings of the National Academy of Sciences* **82(7)**:2192–2195.

**Smith K, Seligman L, Swarup V. 2008.** Everybody share: the challenge of data-sharing systems. *Computer* **41(9)**:54–61.

**Uylings HB, De Brabander J. 2002.** Neuronal changes in normal human aging and Alzheimer's disease. *Brain and Cognition* **49(3)**:268–276.

**Van Ooyen A. 2011.** Using theoretical models to analyse neural development. *Nature Reviews Neuroscience* **12(6)**:311.

**Van Ooyen A, Duijnhouwer J, Remme MW, Van Pelt J. 2002.** The effect of dendritic topology on firing patterns in model neurons. *Network: Computation in Neural Systems* **13(3)**:311–325.

**Van Pelt J, Van Ooyen A, Uylings HB. 2001.** The need for integrating neuronal morphology databases and computational environments in exploring neuronal structure and function. *Anatomy and Embryology* **204(4)**:255–265.

**Waters J, Schaefer A, Sakmann B. 2005.** Backpropagating action potentials in neurones: measurement, mechanisms and potential functions. *Progress in Biophysics and Molecular Biology* **87(1)**:145–170.

**Watson KK, Jones TK, Allman JM. 2006.** Dendritic architecture of the von economo neurons. *Neuroscience* **141(3)**:1107–1112.

**Wearne S, Rodriguez A, Ehlenberger D, Rocher A, Henderson S, Hof P. 2005.** New techniques for imaging, digitization and analysis of three-dimensional neural morphology on multiple scales. *Neuroscience* **136(3)**:661–680.

**Wen Q, Chklovskii DB. 2008.** A cost-benefit analysis of neuronal morphology. *Journal of Neurophysiology* **99(5)**:2320–2328.

**Zandt B-J, Veruki ML, Hartveit E. 2018.** Electrotonic signal processing in AII amacrine cells: compartmental models and passive membrane properties for a gap junction-coupled retinal neuron. *Brain Structure and Function* **223(7)**:3383–3410.

**Zang Y, Dieudonné S, De Schutter E. 2018.** Voltage-and branch-specific climbing fiber responses in purkinje cells. *Cell Reports* **24(6)**:1536–1549.