# Peer review of "Spatially resolved dendritic integration: towards a functional classification of neurons"

_PeerJ, doi:10.7717/peerj.10250_

## Round 0.1 · original submission · Major Revisions

Dear Authors,
Please address the reviewers' concerns and revise your manuscript.

Reviewer 1 ·

Basic reporting

1. Line 40: "Filters" should be "filter", just grammatically.

2. Line 39 -1: If by punctual you mean point-neuron models, these have no dendrites at all. Furthermore, the statement, "dendrites are not passive media that filters electric input" makes it seem
like there is no passive filtering at all, which there is. More generally, the study clearly focuses on active, nonlinear properties of dendrites, but the article would improve by acknowledging the existence of passive, linear properties, and especially by briefly explaining your reasoning for focusing on dendritic spikes.

3. Table 1: Please include brain region for each cell - for instance, hippocampal vs cortical pyramidal cells

4. While I understand that the model has been previously published elsewhere, I believe this study would improve from a slightly more detailed description of it. Could you elaborate on the justification for the probability of spike propagation between compartments being < 1? Is there experimental evidence that in the absence of inhibitory inputs, dendritic spikes fail to propagate once initiated?

Experimental design

1. The list of included cells is heavily biased towards pyramidal cells. Adding some more diversity to the cell types investigated would greatly increase
quality of the study.

2. Do you perform any quality control on morphologies you're using (i.e. check for irregularities in diameter or other artifacts)? Some artifacts are present in morphologies available on NeuroMorpho, and sometimes morphologies from certain labs can be plagued with specific problems. Some of these probably have little impact on the results, but it would be helpful to know if the cells you chose pass any sort of initial testing.

3. It seems as though the dendritic spikes and backpropagating action potentials are treated equivalently here. If this is not the case, please clarify. If it is, however, I do not believe there is sufficient justification. Since spikes generated in the dendrites are mediated by NMDA or voltage-gated calcium channels, and backpropagating action potentials are mediated by sodium channels, this will impact the spiking dynamics, or in the context of this
model the state transition probabilities. I would expect this would also distinguish their impact on energy consumption.

4. What seems to me a shortcoming of the model is that the probability of generating a spike
in one compartment is not influenced by the synaptic activity (subthreshold post synaptic potentials)
in adjacent compartments. Another is the uniformity of dynamics. Different types of spike are generated in different parts of the neuronal arbor. Furthermore, some review, even if cursory, of what has been seen experimentally regarding dendritic spikes in the cell types studied here. For instance, are dendritic spike as prevalent in pyramidal cells as in Purkinje cells?

Validity of the findings

1. How does categorization of cells based on energy consumption compare to their anatomical and
physiological categories?

Additional comments

This study addresses some important, pressing questions in neuroscience: What is the relationship between morphology and dendritic spiking, and what are the energetic requirements? The authors employ interesting and novel methods to explore these questions. I have two general comments that if addressed I believe will greatly improve the manuscript. First, while all models make simplifications, I think it would be worth further addressing what this model ignores and why it ignores it. Second, I think it would be useful to try drawing more of a connection between your results and physiology (e.g. What particular cell types might belong to which energy consumption categories? Or how does the dynamic range of dendrites observed in this study compare to experimentally observed dendritic spike frequencies?)

·

Basic reporting

This manuscript is well written overall with sufficient background context and references provided. However, the following points might be improved for readers to better comprehend the major findings of this manuscript.

1. At the end of the introduction, the authors stated that the analysis of the topological structure of dendritic trees would help the classification of neurons. Here the authors should clarify what it means to classify neurons, in terms of functionality, or types of excitatory/inhibitory synapses, or anatomical regions, or consumption of energy.

2. In the method section, the authors did a great job describing the SIRS model but the readers will understand more easily and intuitively if the equations are provided explicitly.

3. In the method section, it seems that the authors did not specify which software/HPC was used to run these simulations.

Experimental design

1. The authors selected 26 digital reconstructions from NeuroMorpho. However, it is unclear why these 26 neurons were picked. It will make the study more impactful if the authors show that the samples in this study are representative of most neuronal topological structures.

2. The use of network metrics such as closeness centrality is very clever in evaluating the neuronal structure. However in this study based on SIRS model, only excitatory input is considered whereas in reality each neuron is embedded with a combination of inhibitory and excitatory inputs. The coexistence of inhibitory and excitatory inputs, theoretically, could make the spiking properties much more complicated. I am wondering if the authors could provide some insight on how the addition of inhibitory input would affect the classification.

3. SIRS model, in terms of computational cost, is relatively low. In this case, is it possible to test one neuron from each of the three categories by imposing a soma-to-soma connection within an Erdos-Renyl network given a fixed connection probability? On a network level, consumption of energy and closeness centrality would be more functionally relevant.

Validity of the findings

Based on the various centrality and consumption of energy properties, the authors proposed a 3-category classification of neurons. However, if possible, a more thorough functional implication is appreciated to show if this classification is any indication with respect to its anatomical region.

Furthermore, to validate the symmetry of the dendritic modeling, the manuscript will be more impactful to include some electrotonic analysis. For example, dendritic structures from NeuroMorpho could be imported in NEURON or other simulators, where the inward/outward impedance could be easily computed. I understand that at this stage adding in dendritic diameters would add considerable complexity, but this type of validation is necessary to make any functional connection between dendritic topology and firing properties.

Additional comments

The authors proposed a very novel and systematic way of analyzing neuronal dendritic topology by adopting concepts such as the consumption of energy and centrality metrics. In conclusion, the authors classified neurons into three categories based on their dynamic range and firing behavior.

However, in the field of computational neuroscience, multiple simulators are on the market for biophysically realistic modeling of complicated dendritic structure, on both cellular and network levels. With the help of HPCs and parallel computing, multi-compartmental models of neuronal networks have been proposed. Under this circumstance, the insight drawn from a simplified model without dendritic spatial structure or diversified synapses is limited. Therefore several aspects could be improved to make this work more impactful and helpful to the modeling society.

---

## Round 0.2 · accepted · Accept

Congratulations. Your manuscript has been accepted.
It will be processed for production.

Reviewer 1 ·

Basic reporting

no comment

Experimental design

no comment

Validity of the findings

no comment

Additional comments

I'd like to thank the authors for taking my prior comments, as well as those of the other reviewer, seriously and thoroughly addressing each of them. I think the manuscript has greatly improved over the previous draft and recommend it should be accepted for publication.

·

Basic reporting

In the revision, the authors have addressed all of my previous concerns.

Experimental design

In the revision, the authors have addressed all of my previous concerns.

Validity of the findings

In the revision, the authors have addressed all of my previous concerns.

Additional comments

Overall, the revision clarified the major findings and limitations of this work extensively and, hereby, addressed all of my previous concerns.